# Naked mole-rat very-high-molecular-mass hyaluronan exhibits superior cytoprotective properties

Masaki Takasugi [1], Denis Firsanov[1], Gregory Tombline[1], Hanbing Ning[2], Julia Ablaeva[1], Andrei Seluanov [1]✉ & Vera Gorbunova [1]✉

Naked mole-rat (NMR), the longest-living rodent, produces very-high-molecular-mass hyaluronan (vHMM-HA), compared to other mammalian species. However, it is unclear if exceptional polymer length of vHMM-HA is important for longevity. Here, we show that vHMM-HA (>6.1 MDa) has superior cytoprotective properties compared to the shorter HMM-HA. It protects not only NMR cells, but also mouse and human cells from stress-induced cell-cycle arrest and cell death in a polymer length-dependent manner. The cytoprotective effect is dependent on the major HA-receptor, CD44. We find that vHMM-HA suppresses CD44 protein-protein interactions, whereas HMM-HA promotes them. As a result, vHMM-HA and HMM-HA induce opposing effects on the expression of CD44-dependent genes, which are associated with the p53 pathway. Concomitantly, vHMM-HA partially attenuates p53 and protects cells from stress in a p53-dependent manner. Our results implicate vHMM-HA in anti-aging mechanisms and suggest the potential applications of vHMM-HA for enhancing cellular stress resistance.

[1] Department of Biology, University of Rochester, Rochester, NY 14627, USA. [2] Department of Digestive Diseases, The First Affiliated Hospital of Zhengzhou University, Zhengzhou, Zhengzhou, Henan 450052, People's Republic of China. ✉email: andrei.seluanov@rochester.edu; vera.gorbunova@rochester.edu

The longest-living rodent, the naked mole-rat (NMR) (*Heterocephalus glaber*), has a maximum lifespan of more than 30 years, which is fivefold greater than predicted by body mass[1]. NMR does not show increase in mortality rates for at least 18 years[2], and seems to be protected from age-related deterioration such as metabolic decline, diabetes, and osteoporosis[3,4] Moreover, reproductive function of female NMR even increases with age until over 20 years of age[4]. These features indicate that NMR has evolved efficient anti-aging mechanisms. However, although NMR is increasingly appreciated as a model for aging research, how they resist aging processes remains largely unknown. Intriguingly, although basal levels of oxidative stress are higher in NMR cells than in mouse cells[5,6], the age-related increase in oxidative damage is much smaller in the NMR[7]. Also, NMR proteins are better at maintaining their structure and function under oxidative stress[6,7], which could be at least partially explained by high activities of proteasome and autophagy in NMR cells[8,9]. These observations suggest that handling of cellular stress, rather than the suppression of damage itself, is pivotal for the NMR's longevity. Consistent with this idea, we recently found that the same dose of irradiation induces similar DNA damage but less cellular senescence in NMR cells than in mouse cells[10].

In addition to the negligible organismal senescence, the NMR shows extremely high resistance to cancer. NMR cells express a unique INK4 isoform named pALT[INK4a/b], which is upregulated upon a variety of stresses and has a stronger ability to induce cell-cycle arrest than p15[INK4b] or p16[INK4a][11]. It has been also reported that loss of ARF triggers cellular senescence without affecting the expression levels of p16[INK4A], p21[Cip1/Waf1], and p27[Kip1] in NMR cells[12]. These species-specific functions of tumor suppressor genes may be contributing to the cancer resistance in NMR. Another important NMR-specific anti-cancer mechanism is early contact inhibition (ECI)[13–15]. Cultured NMR fibroblasts are hypersensitive to contact inhibition and stop proliferating at relatively low cell density in a hyaluronan (HA)-dependent manner. HA, a linear polysaccharide and a major component of the extracellular matrix, plays a role in supporting tissue structure and regulating cellular signaling pathways depending on its polymer length[15,16]. Dynamic regulation of the amount and polymer length of HA is implicated in diverse biological processes including cell proliferation, cell migration, and inflammation. In healthy tissues, most of HA is of high-molecular-mass ($M >$ 1 MDa) (HMM-HA). In pathological circumstances, significant fragmentation of HA occurs, giving rise to low-molecular-mass HA ($M < 250$ kDa)[15,17]. NMR produces very-high-molecular-mass hyaluronan (vHMM-HA), much longer than HA in other mammalian species[14]. However, it is still not clear whether HA of exceptionally high polymer length is functionally different from regular HMM-HA. One of the mechanisms underlying the polymer length-dependent HA signal transduction is differential clustering of HA receptors triggered by their multivalent interaction with HA[16,18] Multiple studies have shown that low-, but not HMM-HA promotes inflammatory responses[19]. On the other hand, HMM-HA enhances cellular stress resistance, especially against oxidative stress[20–23], although the underlying mechanisms are not well understood. One caveat here is that the concentrations of HA used in these studies were relatively high; more than 100 μg/ml, 1-2 mg/ml in many cases, whereas most organs except dermis and cartilage have HA content of 1-100 μg HA/g wet tissue[17,24].

The observations that HA exhibits polymer length-dependent cytoprotective effect and that long-lived NMR produces vHMM-HA lead to a hypothesis that additional polymer length of NMR-HA confers superior cytoprotection that could contribute to the NMR's longevity. Here, we show that vHMM-HA has superior cytoprotective properties compared to the shorter HMM-HA.

Mechanistically, vHMM-HA and HMM-HA have opposing effects on CD44 protein–protein interactions (PPIs) and CD44-dependent gene expression. As a result, vHMM-HA partially attenuates the p53 pathway and protects cells from stress-induced cell-cycle arrest and cell death.

## Results

**NMR-HA protects cells from oxidative stress via CD44**. To explore the potential cytoprotective effects of NMR-HA, we first examined whether auto/paracrine actions of HA secreted by NMR skin fibroblasts (NSF) protect them from *tert*-butyl hydroperoxidase (tBHP)-induced cell death. tBHP is a widely used oxidative stressor that produces peroxyl/alkoxyl radicals or depletes GSH through metabolization. As expected, HA degrading enzyme, *Streptomyces* hyaluronidase (HAase), decreased the viability of NSF upon 2 days of tBHP-treatment (Supplementary Fig. 1A). In addition, the conditioned medium (CM) of NSF, but not that of mouse skin fibroblasts (MSF), suppressed the cell death of well-characterized human primary lung fibroblasts (IMR90 cells) upon 2 days of tBHP-treatment in a HA-dependent manner (Supplementary Fig. 1B).

HA can confer cytoprotective effect by directly scavenging ROS in the extracellular space or by triggering intracellular cytoprotective signaling pathways. In order to test whether NMR-HA protects cells by enhancing cellular stress resistance rather than by scavenging ROS, we pre-incubated IMR90 cells with 20 μg/ml (physiological concentration in many tissues) of purified NSF-HA or equivalent volume of PBS for 6 h, and then removed HA- or PBS-containing media and treated cells with high-dose tBHP for 1 h. This way HA was not present during tBHP treatment removing its direct ROS scavenging effect. As shown in Fig. 1a, 6-h pre-incubation with NSF-HA was enough to suppress tBHP-induced cell death. Daily repetition of these treatments using low- instead of high-dose tBHP resulted in a NSF-HA-dependent recovery of cell proliferation (Fig. 1b). Without tBHP-treatment, NSF-HA neither promoted cell proliferation nor induced ECI-like cell cycle arrest in IMR90 cells, indicating that NSF-HA is not influencing the cell cycle by itself (Supplementary Fig. 1C). NSF-HA pre-incubation also reduced the number of DNA damage foci after the repetitive low-dose tBHP treatment (Fig. 1c; Wilcoxon test $p = 6.8 \times 10^{-6}$). We then examined the involvement of the two major HA receptors, CD44 and RHAMM, in the cytoprotective effect of NSF-HA. Knockdown of RHAMM did not block the cytoprotective effect of NSF-HA, whereas CD44 siRNA as well as a CD44 neutralizing antibody abrogated its effect (Fig. 1d, e and Supplementary Fig. 2). These results indicate that HA-CD44 signaling is responsible for the cytoprotection by NSF-HA.

**vHMM-HA has superior cytoprotective properties**. To assess whether the exceptional polymer length of NSF-HA contributes to its cytoprotective effect, we pre-incubated IMR90 cells with NSF-HA or the same amount (20 μg/ml) of MSF-HA for 6 h before tBHP-treatment. Majority of NSF-HA was vHMM-HA that has molecular mass of higher than 6.1 MDa, whereas entire MSF-HA was smaller than 6.1 MDa (Fig. 2a), as has been reported previously[14]. Unlike NSF-HA, MSF-HA did not enhance oxidative stress resistance in IMR90 cells (Fig. 2b, c), although the median molecular size of MSF-HA still falls in the class of HMM-HA. To exclude the possibility that this difference is due to the impurities in two HA preparations, we next compared the effects of intact and partially fragmented NSF-HA (fNSF-HA) on cellular stress resistance. For partial fragmentation, NSF-HA was incubated with low concentration of HAase for short period of time, and the reaction was stopped by heat inactivating the enzyme. For control, NSF-HA was heated after mixing with

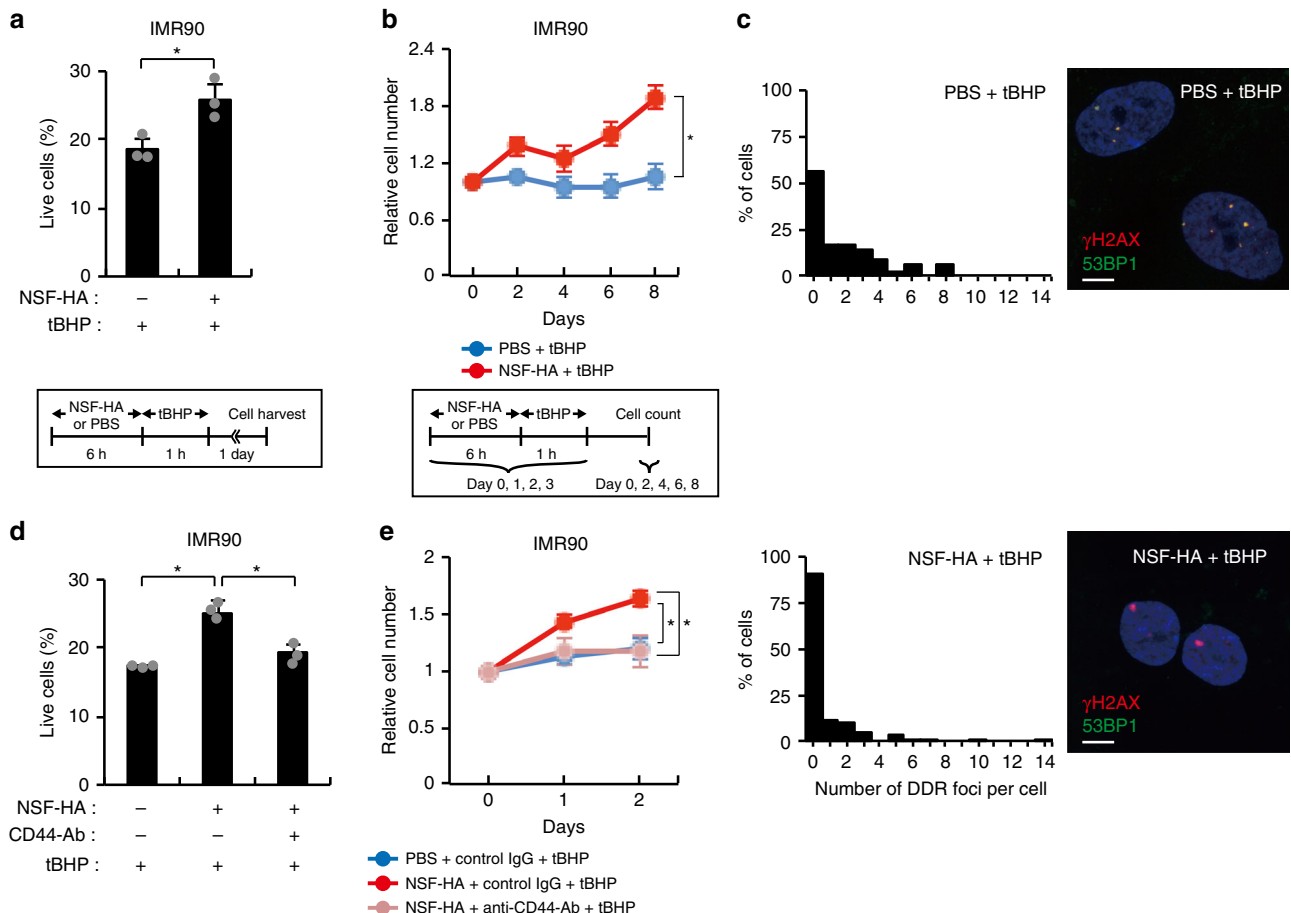

**Fig. 1 The effects of NSF-HA on cellular resistance to oxidative stress. a** NSF-HA suppresses oxidative stress-induced cell death in IMR90 cells. The percentages of live cells were measured by Annexin-V/PI staining 1 day after 1 h of 3 mM tBHP-treatment (*n* = 3). Cells were pre-incubated for 6 h with 20 μg/ml NSF-HA or PBS before tBHP-treatment. **b** NSF-HA suppresses oxidative stress-induced growth arrest in IMR90 cells (*n* = 4). Cells were pre-incubated for 6 h with 20 μg/ml NSF-HA or PBS and then HA was removed and cells were exposed to 200 μM tBHP for 1 h. These treatments were repeated daily for the first 4 days. **c** IMR90 cells were stained with γH2AX and 53BP1 antibodies at day 8 of the experiment represented in (**b**). Histograms show the distributions of the number of γH2AX/53BP1 colocalized foci per cell. For each sample group, experiment was repeated (*n* = 2) and more than 100 cells were counted in total. Scale bars, 20 μm. **d** CD44 antibody blocks the cytoprotective effect of NSF-HA on oxidative stress-induced cell death in IMR90 cells. The percentages of live cells were measured by Annexin-V/PI staining 1 day after 1 h of 3 mM tBHP-treatment (*n* = 3). Cells were pre-incubated for 6 h with 20 μg/ml NSF-HA or PBS with or without CD44 blocking antibody (10 ng/ml) before tBHP-treatment. **e** CD44 antibody blocks the cytoprotective effect of NSF-HA on oxidative stress-induced growth arrest in IMR90 cells (*n* = 4). Cells were pre-incubated for 6 h with 20 μg/ml NSF-HA or PBS with or without CD44 blocking antibody (10 ng/ml) and then HA was removed and cells were exposed to 200 μM tBHP for 1 h on day 0 and day 1. Scale bars, 20 μm. Error bars are presented as mean ± SD values. *$p < 0.05$ [two-tailed *t*-test for (**a**, **b**) and one-way ANOVA with *post-hoc* Dunnett's two-tailed test for (**d**, **e**)].

heat-inactivated HAase. Therefore, control NSF-HA (cNSF-HA) and fNSF-HA should be exactly identical except for the HA polymer length. Although the majority of fNSF-HA retained the molecular mass of higher than 1 MDa, it no longer protected IMR90 cells from tBHP-induced stress (Fig. 2d–f). Note that molecular size distributions of cNSF- and fNSF-HA were unchanged during the incubation with IMR90 cells, indicating that the absence of the cytoprotective effect of fNSF-HA is not due to the degradation of HMM-HA during the experiment (Supplementary Fig. 3A). In addition, cNSF-HA but not fNSF-HA protected against doxorubicin (DXR)- and irradiation-induced cell-cycle arrest in IMR90 cells (Supplementary Fig. 3B, C). MSF were also protected by NSF-HA in a polymer length-dependent manner (Supplementary Fig. 3D, E). Finally, we compared the cytoprotective effect of gel-extracted vHMM-HA (>6.1 MDa) and synthetic hyaluronan (Select-HA^TM) with a uniform molecular mass of 1 MDa. Gel-extracted vHMM-HA protected IMR90 cells from oxidative stress, but 1 MDa HA did

not, even at fivefold higher concentration (Supplementary Fig. 3F-I). These results indicate that vHMM-HA has superior cytoprotective properties over the shorter HMM-HA.

**vHMM- and HMM-HA have opposing effects on CD44 interactome.** The fact that the cytoprotective effect of NSF-HA depends on CD44 and its exceptional polymer length suggests that vHMM-HA and HMM-HA regulate CD44 signals in different ways. To test this notion, we analyzed the effect of HA on CD44 protein-protein interactions. We overexpressed standard form of CD44 in IMR90 cells (Supplementary Fig. 4A) and performed CD44 co-immunoprecipitation assay 6 h after incubating with cNSF-HA, fNSF-HA, or PBS. The majority of endogenous CD44 molecules were of standard form in IMR90 cells, MSF[10], and NSF (Supplementary Fig. 4B, C). CD44 and its interactors were cross-linked with DSP and DTME prior to cell harvest. Immunoprecipitated proteins were then subjected to

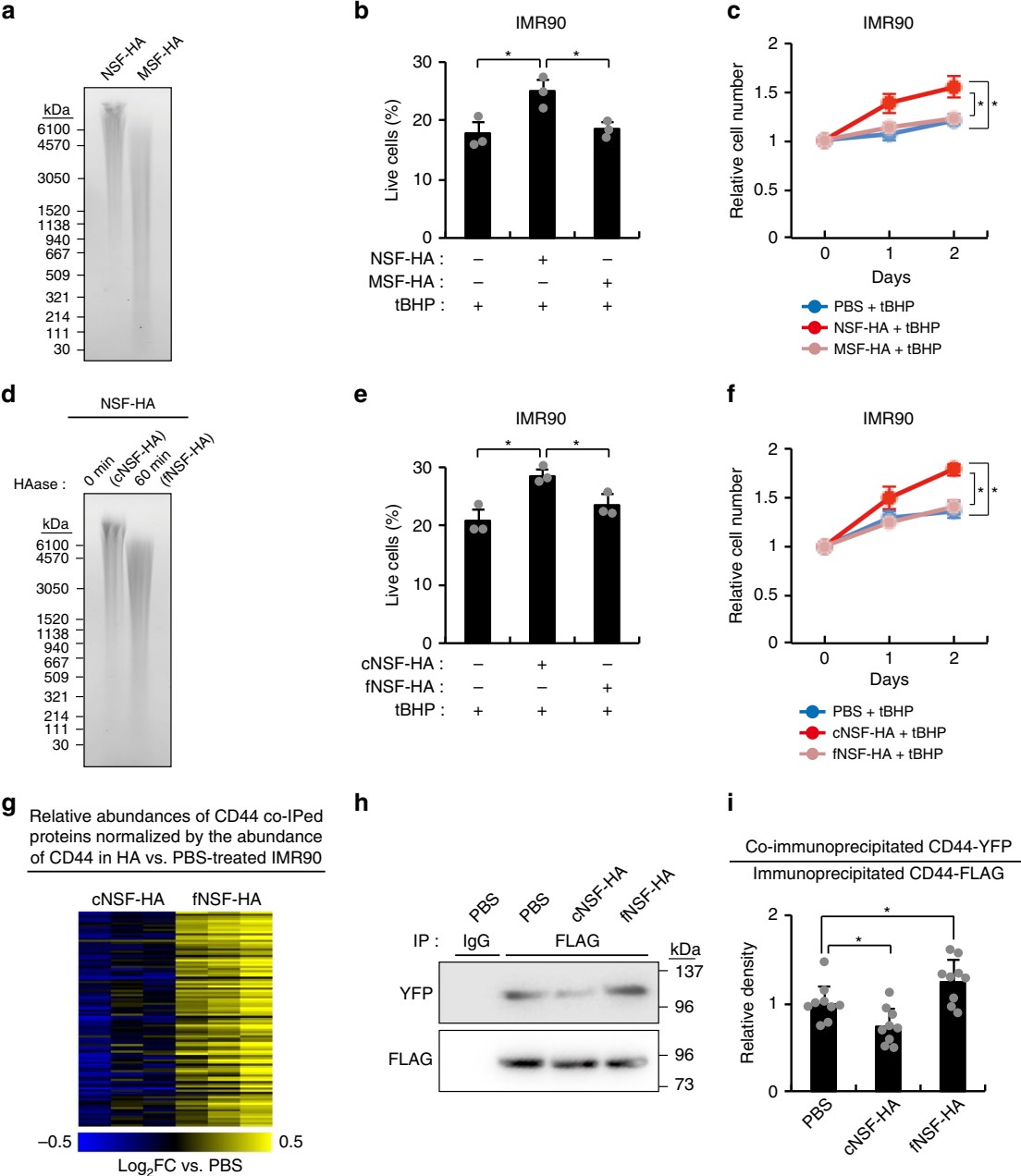

**Fig. 2 vHMM-HA has superior cytoprotective properties. a** Pulse-field gel electrophoresis image of NSF-HA and MSF-HA. The experiment was repeated once with similar result. **b** NSF-HA, but not MSF-HA, suppresses oxidative stress-induced cell death in IMR90 cells. The percentages of live cells were measured by Annexin-V/PI staining 1 day after 1 h of 3 mM tBHP-treatment ($n = 3$). Cells were pre-incubated for 6 h with 20 μg/ml NSF-HA, MSF-HA, or PBS before tBHP-treatment. **c** NSF-HA, but not MSF-HA, suppresses oxidative stress-induced growth arrest in IMR90 cells ($n = 4$). Cells were pre-incubated for 6 h with 20 μg/ml of NSF-HA, MSF-HA, or PBS and then HA was removed and cells were exposed to 200 μM tBHP for 1 h on day 0 and day 1. **d** Pulse-field gel electrophoresis image of cNSF-HA and fNSF-HA. The experiment was repeated once with similar result. **e** cNSF-HA, but not fNSF-HA, suppresses oxidative stress-induced cell death in IMR90 cells. The percentages of live cells were measured by Annexin-V/PI staining 1 day after 1 h of 3 mM tBHP-treatment ($n = 3$). Cells were pre-incubated for 6 h with 20 μg/ml cNSF-HA, fNSF-HA, or PBS before tBHP-treatment. **f** cNSF-HA, but not fNSF-HA, suppresses oxidative stress-induced growth arrest in IMR90 cells ($n = 4$). Cells were pre-incubated for 6 h with 20 μg/ml cNSF-HA, fNSF-HA, or PBS and then HA was removed and cells were exposed to 200 μM tBHP for 1 h on day 0 and day 1. **g** The heatmap shows the CD44-normalized abundances of 99 CD44 co-immunoprecipitated proteins that are differentially associating with CD44 in CD44-overexpressing IMR90 cells. Cells were incubated for 6 h with 20 μg/ml cNSF-HA, fNSF-HA, or PBS. The experiment was done in triplicates. (**h**) Western blots showing the abundances of CD44-FLAG co-immunoprecipitated CD44-YFP in IMR90 cells overexpressing CD44-FLAG and CD44-YFP ($n = 9$). Cells were incubated for 6 h with 20 μg/ml cNSF-HA, fNSF-HA, or PBS. **i** Quantification of Western blots shown in (**h**) by Image J. Error bars are presented as mean ± SD values. *$p < 0.05$ (one-way ANOVA with post-hoc Dunnett's two-tailed test).

quantitative tandem-mass-tag (TMT) mass spectrometry. This identified 290 co-immunoprecipitated proteins with enrichment of at least twofold compared to IgG control (Supplementary Dataset 1). Of those, 109 proteins (38%) have been already reported as CD44-interacting proteins[25], indicating the success of co-immunoprecipitation. We compared the abundances of CD44 co-immunoprecipitated proteins after normalization to CD44 levels and identified 99 proteins to be differentially associating with CD44 in cNSF-HA- and fNSF-HA-incubated IMR90 cells ($p < 0.05$). Strikingly, co-immunoprecipitation efficiencies of all of these proteins were higher in cells incubated with fNSF-HA than those incubated with cNSF-HA (Supplementary Dataset 1). Moreover, when compared to PBS, cNSF-HA reduced the co-immunoprecipitation efficiencies of the majority of these proteins, whereas fNSF-HA mostly enhanced them (Fig. 2g). These results show that CD44 protein-protein interactions are promoted by HMM-HA but are suppressed by vHMM-HA. Next, to clarify the effect of vHMM-HA on protein-based CD44-CD44 interaction, we simultaneously overexpressed CD44-FLAG and CD44-YFP in IMR90 cells and examined their interaction by FLAG co-immunoprecipitation assay. Consistent with our mass-spec data, cNSF-HA reduced the amount of co-immunoprecipitated CD44-YFP, whereas fNSF-HA increased it (Fig. 2h, i). This result suggests that very large vHMM-HA molecules shield CD44 and reduce its interaction with other proteins.

**vHMM- and HMM-HA have opposing effects on gene expression**. Our data imply that vHMM-HA protects cells by enhancing cellular stress resistance via altered CD44 signaling. In support of this notion, pre-incubation with NSF-HA did not reduce intracellular ROS induction after the tBHP-treatment in IMR90 cells (Supplementary Fig. 5A). Also, short HAase treatment following NSF-HA incubation did not impair its cytoprotective effect (Supplementary Fig. 5B, C). This indicates that residual pericellular HA is unlikely to be important for the cytoprotection. Although internalized HA may be involved, our data nevertheless suggest that intracellular signals downstream of CD44 play a key role in the cytoprotection by vHMM-HA.

Therefore, in order to further investigate the mechanisms of the cytoprotective effect, we first aimed to clarify how vHMM-HA affects transcriptional signature in stressed cells. We compared the transcriptomes of IMR90 cells 6 h after starting 1-h low-dose tBHP-treatment and identified 745 genes that are differentially expressed between cNSF-HA- and fNSF-HA pre-incubated cells ($q < 0.05$; hereafter referred to as HA polymer length-dependent genes). Expression levels of most of these genes were oppositely regulated by cNSF-HA and fNSF-HA (significant negative correlation; Spearman correlation test, $p < 2.2 \times 10^{-16}$) (Fig. 3a), as is the case with CD44 protein-protein interactions. Thus, although CD44 PPIs were investigated under the condition of CD44 overexpression that can affect the results, our proteomic and transcriptomic analyses both indicated contrasting effects of cNSF-HA and fNSF-HA. Gene ontology analysis[26] of the HA polymer length-dependent genes showed the enrichment of genes encoding the regulators and interactors of p53 (Fig. 3b). Next, we conducted motif-based transcription factor binding site (TFBS) enrichment analysis[27] and found that targets of CD44-regulated transcription factors such as ELK1[28] and EGR1[29] were over-represented in the HA polymer length-dependent genes (Fig. 3c). Moreover, the most enriched transcription factor targets, namely, the targets of HIF1α, have been shown to overlap significantly with the targets of CD44-ICD (CD44 intra-cytoplasmic domain) that is generated by proteolysis of CD44[30]. Indeed, CD44-ICD levels were lower in fNSF-HA-incubated cells (Supplementary

Fig. 6), and there was an enrichment of CD44-ICD binding motif (CCTGCG) in the 500 bp promoter regions of the HA polymer length-dependent genes (Fig. 3d).

All of these TFBS enrichments become more prominent when the analysis was restricted to the HA polymer length-dependent genes encoding regulators and interactors of p53 (Fig. 3d), implying a major role of the p53 pathway in the CD44-dependent cytoprotective effect of NSF-HA. Consistent with these observations, there was a significant enrichment of curated high-confident p53 target genes[31] in the HA polymer length-dependent genes (Fisher's exact test, $p = 1.7 \times 10^{-6}$) (Supplementary Fig. 7A). Notably, cNSF-HA pre-incubation suppressed the expression of most of the HA polymer length-dependent p53 target genes (Supplementary Fig. 7A, B). With that being said, majority of the p53 target genes respond similarly to the oxidative stress in cNSF-HA, fNSF-HA, and PBS-indubated cells (Supplementary Fig. 7C). Cytoplasmic and nuclear levels of p53 were also unaffected by HA (Supplementary Fig. 7D–G). Target genes of CD44-ICD and CD44-regulated transcription factors were not enriched in the HA polymer length-dependent p53 target genes (Fig. 3d), suggesting that the transcriptional modulation of HA polymer length-dependent p53 target genes is a consequence of altered expression of genes encoding regulators and interactors of p53 that are regulated by CD44-regulated transcription factors and CD44-ICD.

Interestingly, we found that early stress responsive gene expression changes induced by tBHP and irradiation[32] show positive correlation with the gene expression changes induced by cNSF-HA pre-incubation in the HA polymer length dependent genes (Spearman correlation test, $p < 2.2 \times 10^{-16}$ for tBHP and $p = 3.2 \times 10^{-5}$ for irradiation, Supplementary Fig. 7H). This is surprising since p53 target genes as a whole were predominantly upregulated upon these early stress responses (Supplementary Fig. 7A). These results suggest that cNSF-HA pre-incubation potentiates a subset of early stress-response that is especially associated with p53 suppression.

Importantly, transcriptomic analysis of non-stressed IMR90 cells incubated with cNSF-HA, fNSF-HA, and PBS showed that the expression changes of most of the HA polymer length-dependent genes were already induced before tBHP-treatment (Fig. 3e). Other genes showed much less HA polymer length-dependent effects in these cells (Fig. 3f). RT-qPCR analysis of non-stressed control and CD44 knocked down IMR90 cells confirmed that the polymer length-dependent effects of HA on the expression of p53 target genes (*RRM2B*[33], *PRRX2*[25], and *PYCARD*[34]) are mediated by CD44 (Supplementary Fig. 8). Finally, in order to further confirm the role of CD44 signaling in the regulation of HA polymer length-dependent genes (and to complement the proteomic analysis described in the previous section), we performed transcriptomic analysis on mock- and CD44-overexpressing IMR90 cells under non-stress conditions. As expected based on the proteomic analysis above, transcriptional effect of CD44 overexpression on the HA polymer length-dependent genes was largely suppressed by incubating cells with cNSF-HA. In the HA polymer length-dependent genes, expression changes induced by CD44-overexpression in PBS-treated IMR90 cells and those induced by cNSF-HA-treatment in CD44-overexpressing IMR90 cells showed a significant negative correlation (Spearman correlation test, $p < 2.2 \times 10^{-16}$) (Fig. 3g). Taken together, these data show that vHMM-HA antagonizes the transcriptional effect of HMM-HA and CD44 and thereby changes the expression of interactors and regulators of p53 that are potentially under control of CD44-related transcription factors, resulting in reduced expression of a subset of p53 target genes.

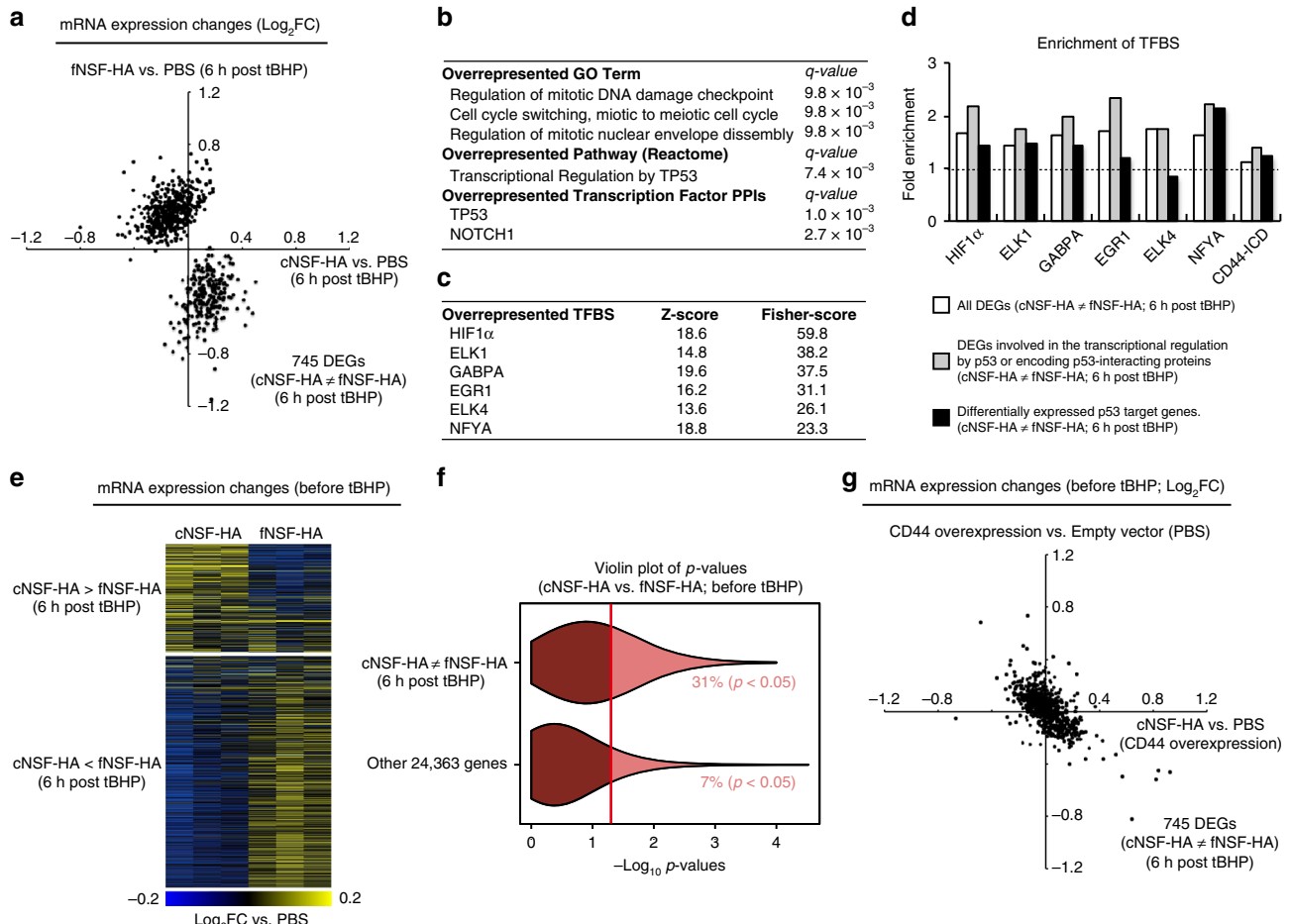

**Fig. 3 Transcriptome analyses of NSF-HA-treated IMR90 cells. a** Genes differentially expressed in IMR90 cells upon treatment with cNSF-HA or fNSF-HA (HA polymer length-dependent genes). Cells were pre-incubated for 6 h with 20 μg/ml cNSF-HA, fNSF-HA, or PBS and then HA was removed and cells were exposed to 200 μM tBHP for 1 h. Cells were collected 6 h after starting the tBHP-treatment. **b** Enrichment of GO Term, Reactome pathway, and transcription factor protein-protein interactions (PPIs) in the HA polymer length-dependent genes. **c, d** Enrichment of transcription factor binding sites (TFBS) in the HA polymer length-dependent genes. White bars represents the enrichment of TFBS in all HA polymer length-dependent genes. Gray bars represents the enrichment of TFBS in HA polymer length-dependent genes that are involved in the transcriptional regulation by p53 or encoding p53-interacting proteins. Black bars represents the enrichment of TFBS in HA polymer length-dependent p53 target genes. **e** Differential expression levels of the HA polymer length-dependent genes (defined in the presence of tBHP) in IMR90 cells incubated for 6 h with 20 μg/ml cNSF-HA, fNSF-HA, or PBS. **f** The violin plot represents the distribution of p-values of a two-tailed t-test comparing the gene expression levels of the HA polymer length-dependent genes and HA-polymer length-independent genes (defined in the presence of tBHP) between cNSF-HA- and fNSF-HA-incubated IMR90 cells. Cells were incubated for 6 h with 20 μg/ml cNSF-HA or fNSF-HA without further tBHP-treatment. **g** Differential expression levels of the HA polymer length-dependent genes (defined in the presence of tBHP) in control and CD44-overexpressing IMR90 cells incubated for 6 h with 20 μg/ml cNSF-HA or PBS.

**vHMM-HA protects cells via attenuating p53**. We next sought to clarify whether vHMM-HA and HMM-HA differentially affect the status of p53 itself. We examined the major p53 N-terminus phosphorylation sites and found that p53 Ser9 was less phosphorylated in cNSF-HA-incubated cells than in fNSF-HA-incubated cells (Fig. 4a, b and Supplementary Fig. 9). To test whether p53 is actually involved in the cytoprotective effect of NSF-HA, we knocked down p53 in IMR90 cells using a validated siRNA[35]. IMR90 cells in which p53 was knocked down were still sensitive to tBHP, but were no longer protected by NSF-HA pre-incubation (Fig. 4c–e). Since siRNA cannot completely silence the target gene, we next conducted the same experiment using wild-type and p53-knockout MSF[36]. Similar to IMR90 cells, p53-knockout MSF were still sensitive to tBHP but were not protected by NSF-HA pre-incubation (Fig. 4f–h). Based on these results, we conclude that cytoprotective effect of NSF-HA is mediated at least partially via attenuating the p53 pathway.

**Discussion**

Although extracellular matrix constitutes a major part of the body, its relationship with aging remains largely unexplored. HA, a linear polysaccharide and a major component of the extracellular matrix, has very high molecular mass in the longest-living rodent, the NMR, but it was unclear whether exceptional polymer length of vHMM-HA contributes to the phenotypes of the NMR. Here, we demonstrated that vHMM-HA (>6.1 MDa) is functionally different from shorter high-molecular-mass HA and that vHMM-HA exhibits distinct cytoprotective effect on NMR, mouse, and human cells in a p53-dependent manner. Our data collectively support a model in which vHMM-HA suppresses the protein-protein interactions and the signalling by CD44, resulting in altered expression of interactors, regulators, and targets of p53, and thereby exhibits distinct cytoprotection through modulating the p53 pathway (Fig. 5). Our findings suggest that vHMM-HA contributes to longevity of the NMR and highlight the potential application of vHMM-HA as a cytoprotective molecule.

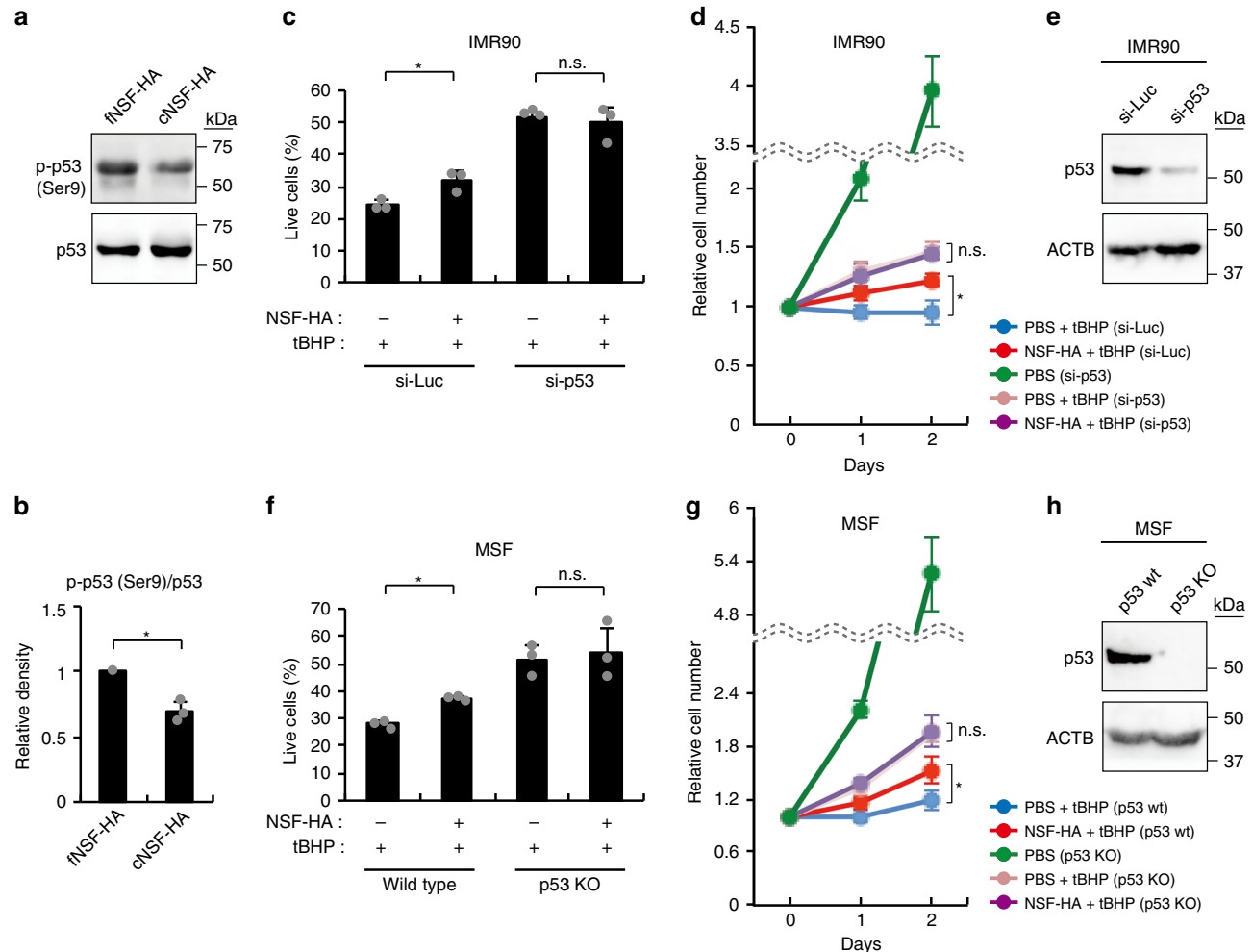

**Fig. 4 Cytoprotective effect of NSF-HA depends on p53. a** Western blots showing p-p53-Ser9 levels in IMR90 cells treated with 20 μg/ml cNSF-HA or fNSF-HA for 6 h. **b** Quantification of Western blots shown in (**a**) by Image J. Averages are of three biological replicates. For each biological replicate, immunoblotting was performed three times and averaged. **c** p53-knockdown abrogates the cytoprotective effect of NSF-HA on oxidative stress-induced cell death in IMR90 cells. The percentages of live cells were measured by Annexin-V/PI staining 1 day after 1 h of 3 mM tBHP-treatment ($n = 3$). Cells were pre-incubated for 6 h with 20 μg/ml NSF-HA or PBS before tBHP-treatment. siRNA was transfected 2 days before the first tBHP-treatment. **d** p53-knockdown abrogates the cytoprotective effect of NSF-HA on oxidative stress-induced growth arrest in IMR90 cells ($n = 4$). Cells were pre-incubated for 6 h with 20 μg/ml NSF-HA or PBS for 6 h and then HA was removed and cells were exposed to 200 μM tBHP for 1 h on day 0 and day 1. siRNA was transfected 2 days before day 0. **e** Confirmation of p53-knockdown in IMR90 cells by Western blot. Cells were collected 2 days after p53 or control siRNA transfection. Immunoblotting was repeated once with similar result. **f** p53-knockout abrogates the cytoprotective effect of NSF-HA on oxidative stress-induced cell death in MSF. The percentages of live MSF were measured by Annexin-V/PI staining 1 day after 1 h of 1.5 mM tBHP-treatment ($n = 3$). Cells were pre-incubated for 6 h with 20 μg/ml NSF-HA or PBS before tBHP-treatment. **g** p53-knockout abrogates the cytoprotective effect of NSF-HA on oxidative stress-induced growth arrest in MSF ($n = 4$). Cells were pre-incubated for 6 h with 20-μg/ml NSF-HA or PBS for 6 h and then HA was removed and cells were exposed to 150-μM tBHP for 1 h on day 0 and day 1. **h** Confirmation of p53-knockout in MSF by Western blot. Immunoblotting was repeated once with similar result. Error bars are presented as mean ± SD values. *$p < 0.05$ (two-tailed $t$-test).

It should be noted, however, that our mechanistic investigations were largely restricted to human cells, as the clinically relevant model. Genes involved in the mechanisms mediating the protective effect of vHMM-HA can be different among species. For example, there are considerable differences in the genes regulated by p53 between human and mouse[37]. Not surprisingly, amino acid sequences of p53 are different among human, mouse, and NMR[38], although no functional differences have been reported between NMR and the two other species. Another important limitation of our study is that we only focused on the distinctive cytoprotective effect of vHMM-HA. Other functions may be also different between vHMM-HA and shorter HMM-HA. In addition, while CD44-p53 signals play a critical role in the cytoprotective effect of vHMM-HA, other HA-binding proteins,

such as versican[39], as well as other aspects of CD44, such as formation of molecular complexes[40] and picket fence[41], may be involved in the cytoprotective effect or other function of vHMM-HA. Although these issues are beyond the scope of this study, our proteomic and transcriptomic data might be useful resources for addressing these questions.

In the current study, we mostly used naturally produced HAs that have considerable polydispersity, and thus there could be functional heterogeneity even within vHMM-HA and HMM-HA. However, based on the results of our experiments using gel-extracted vHMM-HA and 1 MDa Select-HA™, we can conclude that vHMM-HA larger than 6.1 MDa has superior protective effect compared to 1 MDa HA or MSF-HA that consists of HMM-HA smaller than 6.1 MDa. The conclusion that NSF-HA

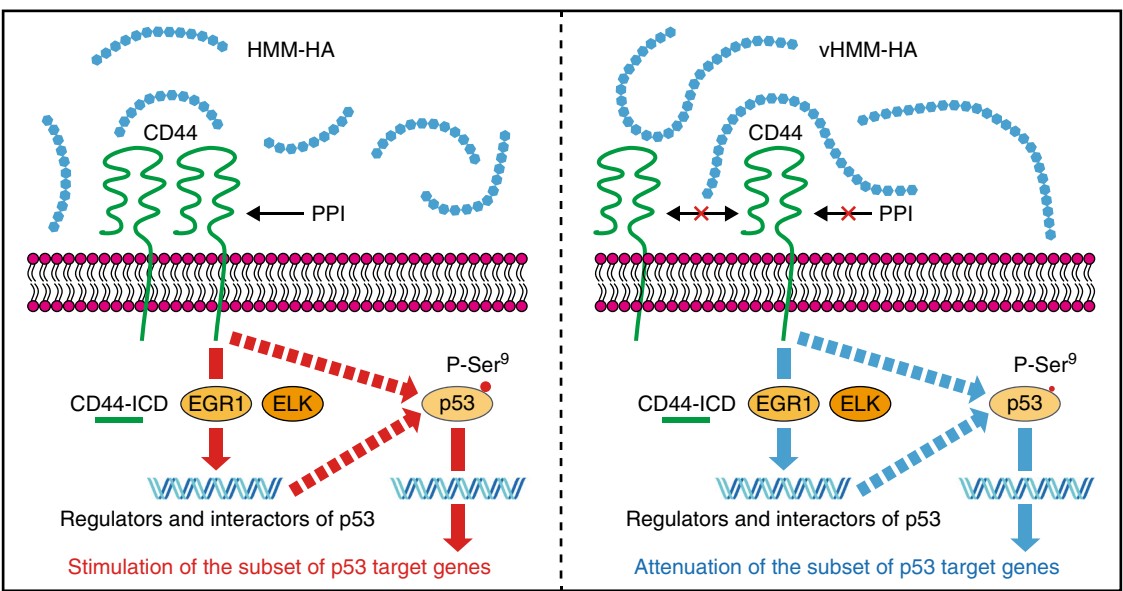

**Fig. 5 Schematic model of the mechanism of protective effect of vHMM-HA.** Our data indicate that vHMM-HA attenuates CD44 protein-protein interactions (PPI), whereas HMM-HA promotes them. Concomitantly, vHMM-HA suppresses CD44-dependent gene expression, including those regulated by CD44-dependent transcription factors such as ELK, EGR1, and CD44-ICD. vHMM-HA and HMM-HA thereby induce opposing effects on the expression of CD44-dependent genes, which are also associated with the p53 pathway. As a result, vHMM-HA partially attenuates p53 and its target genes, and protects cells in a p53-dependent manner.

has superior cytoprotective effect compared to MSF-HA is also valid. These conclusions suggest the contribution of vHMM-HA to the NMR stress resistance and longevity.

Although vHMM-HA partially attenuates the activity of tumor-suppressor p53, the NMR is nevertheless extremely resistant to carcinogenesis. One explanation is that the vHMM-HA-induced partial attenuation of p53 may not necessarily compromise its tumor suppressor activity. Indeed, as a whole, p53 target genes respond very similarly to oxidative stress in PBS- and HA-incubated cells. Differential transcriptional modulation of a subset of p53 target genes by cNSF-HA and fNSF-HA could be a consequence of altered expression of genes encoding regulators and interactors of p53 that are regulated by CD44-regulated transcription factors and CD44-ICD.

Our observation that at physiological levels vHMM-HA better protects human cells from stress than the shorter HMM-HA has significant clinical implications considering the broad medical applications of HA[42]. Manufacturing higher molecular mass HA might lead to the improvement of HA-based medical products.

## Methods

**Cell culture**. NSF and MSF were isolated in our lab[43] from the skin of young adult NMRs and C57BL/6J mice, respectively. Skin fibroblasts were isolated from skin samples collected from euthanized animals by mincing and digesting them with 0.14 Wunch units/ml Liberase TM (Roche Diagnostics) in DMEM/F12 at 37 °C for 60 min. All animals were maintained in accordance with the regulations designated and approved by the University of Rochester Committee on Animal Resources (UCAR), which adheres to FDA and NIH animal care guidelines and reviews all animal protocols prior to approval. IMR90 human lung fibroblasts were from the Coriell Institute for Medical Research. HEK293T cells were from ATCC. NSF, MSF, and IMR90 cells were cultured in EMEM supplemented with 15% FBS. NSF were cultured at 32 °C, 4.8% $CO_2$, 2.8% $O_2$. MSF and IMR90 cells were cultured at 37 °C, 5% $CO_2$, 4% $O_2$. The passage numbers of NSF, MSF, and IMR90 cells were less than 20, 5, and 40, respectively. In IMR90 cells, the sequence of all p53 exons was checked using our RNA-Seq data and was found to have no mutation. Cell counting was done using gridded culture dishes.

**Apoptosis assay**. Apoptotic cells were quantified using Annexin-V Apoptosis Detection Kit (eBioscience) following the manufacturer's instructions. After staining, cells were analyzed with LSR-II flow cytometer (BD Biosciences).

**HA preparation**. For HA purification, conditioned media were first mixed with proteinase K solution (final concentrations of 1 mM Tris-Cl pH 8.0, 2.5 mM EDTA, 10 mM NaCl, 0.05% SDS, 1 mg/ml proteinase K) and incubated at 55 °C for 4 h. Following protein digestion, media were extracted with saturated phenol-chloroform-isoamyl alcohol (Sigma). HA was precipitated with ethanol and centrifugation (4,000 × g for 45 min). HA pellet was dissolved in PBS and then extracted with 1/100 volume of Triton-X114[44]. After Triton-X114 extraction, HA was precipitated again with ethanol. Finally, HA pellet was washed with 70% ethanol and dissolved in PBS. For gel extraction of vHMM-HA, NSF-HA was run on 0.7% low-melting agarose (Agarose-LM plaque; Nacalai Tesque) in TBE buffer. Marker lane was cut off from the gel and stained as described below. NSF-HA-containing gel above 6.1 MDa marker band was cut off and vHMM-HA was recovered from the gel using thermostable agarase (Nippon gene) following manufacturer's instructions. Recovered vHMM-HA was then extracted with PCI and Triton-X114 and precipitated with ethanol as described above. Concentration of HA was measured by the carbazole assay[45]. *Streptomyces* HAase (Sigma) was used for the degradation of HA in the conditioned media and for the partial fragmentation of HA. HA was partially fragmented by incubating with 0.5 U/ml of HAase at 37 °C for 60 min, followed by heat inactivation at 95 °C for 10 min. Select-HA™ was from Echelon Biosciences. For the blockage of HA/CD44 interactions, a CD44 neutralizing antibody (Clone 2C5; R&D) was added to the culture media at a concentration of 10 ng/ml.

**RNA extraction, reverse transcription, and real-time qPCR**. Total RNA was extracted with RNAiso Plus (Takara) and reverse transcribed into cDNA with random hexamers using PrimeScript reagent kit (Takara). Real-time qPCR was performed using TB Green™ Premix Ex Taq II™ (Takara) with Step ONE Plus Real time PCR system (Life Technologies). The primers used in this study are as follows: *ACTB* forward, AGATCAAGATCATTGCTCCTCCTG; *ACTB* reverse GCCGGACTCGTCATACTCCT; *CD44* forward, TGGTGAACAAGGAGTCGTCA; *CD44* reverse, ACACCCCAATCTTCATGTCC; *PRRX2* forward, AGGTGCCTACGGTGAACTGA; *PRRX2* reverse, CTGCCCCCTTTTCTATTGCT; *PYCARD* forward, TGACGGATGAGCAGTACCAG; *PYCARD* reverse, CAGGCTGGTGTGAAACTGAA; *RHAMM* forward, AGCAAGAAGGCATGGAGATG; *RHAMM* reverse, CCCTCCAGTTGGGCTATTTT; *RRM2B* forward, GCCAGGACTCACTTTTTCCA; *RRM2B* reverse, TCCCTGACCCTTTCTTCTGA.

**Pulse-field gel electrophoresis**. Purified HA was mixed with sucrose solution (final concentration of 333 mM) and loaded to a 0.4% SeaKem Gold agarose gel (Lonza). HA-Ladders (Hyalose) were run alongside the samples. Samples were run 12 h at 4 °C at 75 V with a 1–10 running ratio in TBE buffer using CHEF-DRII system (Bio-Rad). After the run, the gel was stained with 0.005% (w/v) Stains-All (Santa Cruz) in 50% ethanol overnight. Then the gel was washed twice with 10% ethanol, exposed to light to decrease background, and photographed with ChemiDoc Imaging System (Bio-Rad).

**Immunofluorescence staining**. Cells were washed with PBS and fixed with 4% formaldehyde at RT for 15 min. After fixation, cells were washed with 3% BSA/PBS and permeabilized with 0.5% Triton X-100/PBS at RT for 20 min. Next, cells were washed with 3% BSA/PBS and blocked in blocking buffer (10% FBS and 1% BSA in PBS) at RT for 1 h. Then cells were incubated with γH2AX (JBW301; Millipore; 1:500) and 53BP1 (ab172580; Abcam; 1:500) antibodies in the blocking buffer at 4 °C for overnight and washed with PBST. Finally, cells were incubated with goat anti-mouse IgG Alexa Fluor 488 and goat anti-rabbit IgG Alexa Fluor 568 (Thermo Fisher Scientific; 1:1000) in the blocking buffer at RT for 1 h, washed with PBST, and mounted in Vectashield containing DAPI (Vector Laboratories). Images were acquired using a Leica SP5 confocal microscope.

**Establishment of CD44-overexpressing IMR90 cells**. CD44- and CD44-FLAG-overexpressing IMR90 cells were established by viral transduction and 1 μg/ml puromycin selection. CD44-FLAG-overexpressing IMR90 cells were further transduced with retroviral vector carrying CD44-YFP and were selected by FACS (Sony SH800). Retroviral supernatant was prepared by co-transfecting HEK293T cells with VSV-G, pUMVC, and pBabe-puro vector. Empty and CD44s expression pBabe vectors were from Addgene (#1764 and #19127). CD44s was cloned into pcDNA3.1-ZNF598-TEV-3xFLAG (#105690, Addgene) and pcDNA3-YFP (#13033, Addgene) to generate CD44-FLAG (C-terminus) and CD44-YFP (C-terminus), respectively. CD44-FLAG and CD44-YFP were then cloned back into pBabe vector for virus production.

**Immunoprecipitation**. Cells were incubated with 20 μg/ml cNSF-HA, fNSF-HA, or equivalent volume of PBS for 6 h, and then the media were removed and cells were cross-linked in PBS containing 500 μM DSP and 500 μM DTME (Thermo Fisher Scientific) for 30 min at RT. Cross-linking was quenched by incubation with PBS containing 20 mM Tris-Cl (pH 7.4) and 5 mM L-Cysteine (Sigma). Cells were harvested in RIPA buffer (50 mM Tris pH 7.4, 150 mM NaCl, 1% NP-40, 0.5% deoxycholic acid, 0.1% SDS) supplemented with complete protease inhibitor cocktail (Sigma). After cell lysis, the buffer was exchanged with 0.05% TBST using Vivaspin column (MWCO = 10 kDa) (GE Healthcare). Immunoprecipitation was done from 200 μg of lysates with 6 μg of anti-CD44 antibodies (3 μg of Abcam anti-CD44 antibody (Cat. ab157107) and 3 μg of Proteintech anti-CD44 antibody (Cat. 15675-1-AP)), normal rabbit IgG (Cell Signaling Technology), anti-FLAG antibody (M2; sigma), or normal mouse IgG (CST) using Dynabeads Protein G magnetic beads (Thermo Fisher Scientific). Antibodies were cross-linked to the beads with DMP (Thermo Fisher Scientific) prior to the immunoprecipitation. Immunoprecipitants were eluted by incubating with RIPA buffer supplemented with 50 mM DTT for 20 min at 70 °C.

**Mass spectrometry**. CD44-immunoprecipitants were trypsin-digested and purified using S-Trap column (Protifi). Peptides were sent to the Mass Spectrometry Resource Lab of the University of Rochester for 10-plex TMT labeling and mass spectrometry. Samples were resolved by nanoelectrospray ionization on an Orbitrap Fusion Lumos MS instrument. Peptide assignments were made using Proteome Discoverer and Sequest and MS3 ions were used for quantifying protein abundances.

**Transcriptome analysis**. RNA was extracted using Trizol (Invitrogen). RNA samples were sent to the Genomics Research Center of the University of Rochester for RNA-Seq. The TruSeq RNA Sample Preparation Kit V2 (Illumina) was used for library construction. The libraries were hybridized to the Illumina single end flow cell and amplified using the cBot (Illumina) at a concentration of 8 pM per lane. Single end reads of 100 nt were generated for each sample. The sequencing was performed using the Illumina high-throughput HiSeq™ 2500. Normalization and differential expression analysis were carried out using DESeq2[46]. Ontology and pathway analyses were performed using Enrichr program[26]. TFBS enrichment analysis was done using oPOSSUM program[27]. The parameters used in the oPOSSUM were as follows: conservation cut-off = 0.4, matrix score threshold = 85%, analyzed sequences = 2 kb upstream of transcription start sites, Z-score cut-off = 10, Fisher score cut-off = 7.

**Intracellular ROS quantification**. Intracellular ROS levels were quantified using the fluorescent probe $H_2DCFDA$. Cells were pre-incubated for 6 h with 20 μg/ml of NSF-HA or equivalent volume of PBS, then with 5 μM $H_2DCFDA$ in HBSS for 30 min, and then exposed to 200 μM tBHP for 1 h, and washed with HBSS. DCF fluorescence was measured with excitation at Ex485/Em530 nm and was normalized to cell number.

**Western blot**. Whole cell lysates were prepared by lysing cells in lysis buffer (20 mM Tris pH 7.5, 250 mM NaCl, 1% N-lauryl sarcosine, 10 mM NaF, 2 mM sodium orthovanadate, 1 mM β-glycerophosphate) supplemented with cOmplete protease inhibitor cocktail (Sigma). Nuclear and cytoplasmic lysates were prepared using the Nuclear Extract Kit (abcam, ab113474). Lysates and immunoprecipitates were denatured at 95 °C (70 °C for CD44 detection) for 10 min in laemmli buffer.

Proteins were separated by SDS-PAGE, transferred onto nitrocellulose membranes, blocked with 5% BSA or 5% milk in TBST for 1 h, and probed with the following primary antibodies: anti-β-actin (sc-47778; Santa Cruz), anti-CD44 (ab157107; abcam), anti-FLAG (M2; sigma), anti-GFP (for YFP detection) (ab6556; abcam), anti-H3 (#9715; CST), anti-p53 (human) (10442-1-AP; Proteintech), anti-p53 (mouse) (ab26; Abcam), anti-phospho-p53 (Ser-9) (#9288; CST), anti-phospho-p53 (Ser-15) (#9286; CST), anti-phospho-p53 (Ser-46) (#2521; CST), and anti-phospho-p53 (Thr-81) (#2676; CST). All primary antibodies were diluted to 1 μg/ml. After washing with TBST, membranes were incubated with secondary antibodies (Sigma) (1:1000) for 1 h at room temperature (RT), washed again with TBST, and visualized with enhanced chemiluminescence reagents. Uncropped gel images are provided in Supplementary Fig. 10.

**RNAi**. RNAi was performed by the transfection of siRNAs using RNAiMAX transfection reagent (Thermo Fisher Scientific). The final concentration of siRNA was 50 nM. The sequences of the siRNA oligos were as follows. si-Luciferase: CGUACGCGGAAUACUUCGAtt. si-CD44: GAACGAAUCCUGAAGACAUC Utt[47]. si-RHAMM: GCUAGAUAUUGCCCAGUUAtt[48]. si-p53: AAGACUCCAG UGGUAAUCUACtt[35].

**Statistics and reproducibility**. Error bars are presented as mean ± SD values. Two-tailed Student's $t$-test and one-way ANOVA with post-hoc Dunnett's two-tailed test were used to assess statistical significance, unless otherwise indicated. Exact $p$ values are provided in the Source Data file. All replicates are biological.

**Reporting summary**. Further information on research design is available in the Nature Research Reporting Summary linked to this article.

## Data availability
Source Data underlying Figs. 1a, b, d, e, 2b, c, e, f, i, 4b–d, f, g and Supplementary Figs. 1A–C, 2A–H, 3B–E, G–I, 4A, B, 5A–C, 6B, 7E, G, 8A–F, 9B are provided as a Source Data file. RNA-Seq data have been deposited in the Gene Expression Omnibus (GEO) database under the accession number GSE116759 [https://www.ncbi.nlm.nih.gov/geo/query/acc.cgi?acc=GSE116759]. Mass spectrometry data have been deposited in the ProteomeXchange with identifier PXD013445 [https://www.ebi.ac.uk/pride/archive/projects/PXD013445].

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

## Acknowledgements

This work was supported by the US National Institute of Health (NIH) grants to V.G. and A.S. M.T. was partially supported by Japan Society for the Promotion of Science (JSPS) and the Uehara memorial Foundation.

## Author contributions

M.T., A.S., and V.G. conceived and designed the study. M.T., D.F., G.T., H.N., and J.A. performed the experiments. M.T., A.S., and V.G. wrote the manuscript.

## Competing interests

The authors declare no competing interests.
