## [Peer Review File · Nature Communications]

Reviewers' comments:

Reviewer #1 (Remarks to the Author):

This paper describes a remarkably thorough set of experiments to define the route by which vHMW-HA from the naked mole rat (NMR) protects mammalian cells (of mouse, human or NMR origin) from a variety of toxic stresses. The authors discovered those protective properties, and here have made substantial progress in pursuit of their underlying mechanistic basis.

The key conclusions are (1.) that vHMW-HA (but not HMW-HA) masks the major HA receptor, CD44, and thus obstructs its protein-protein interactions (based on pull-down proteomics after cross-linking); (2.) that transcriptional effects of vHMW-HA are mediated via suppression of p53 target genes (demonstrated by analyses of transcriptomic data). In both cases, ample and appropriate controls were included, strengthening the conclusions drawn while excluding most alternative interpretations.

These are very important studies, because they indicate likely routes for stress-protective interventions that in principle could be quite effectively applied to humans (although they remain to be designed) and used prophylactically.

The statistical analyses appear valid, and actually are more sophisticated (and appropriate to the data being compared) than I see in most publications. While replications of experiments are rarely included when publishing scientific reports, the fact that very consistent results were obtained by multiple, independent approaches, by single-point assays vs. time courses, and in different cell lines (human IMR90 cells and mouse skin fibroblasts), indicates a high likelihood of reproducibility in other laboratories. Methodological details are certainly sufficient to enable replication.

The paragraphs added to the Discussion will be appreciated by many readers. Supplementary Figure 6 could serve as a graphical abstract; alternatively, it could be moved to the main body of the paper.

Robert J. Shmookler Reis

Reviewer #2 (Remarks to the Author):

This is a revised manuscript from Takasugi et al. that is examining the potential mechanisms of longevity and cancer resistance mediated by hyaluronan (HA) in the naked mole rat (NMR). They demonstrate that NMR skin fibroblasts (NSF) are more resistant to oxidative stress-mediated cell death than mouse skin fibroblasts (MSF). The effect of HA was not from direct scavenging of reactive oxygen species (ROS) as preincubation with NMR-HA protected later exposure to oxidative stress. Blockade of CD44 by an antibody abrogated the effect of NSF-HA. Comparison of very high molecular mass HA (vHMM-HA) to HMM-HA demonstrated a lower oxidative stress resistance for HMM-HA in human fetal fibroblasts (IMR90 cells). Partial fragmentation of vHMM-HA abrogated its protective properties. The authors respond to the criticism of a lack of mechanistic insight by evaluating the protein:protein interactions of CD44. After overexpression of CD44 in IMR90 cells with and without incubation with NSF-HA, fragmented NSF-HA or PBS, CD44 IP and proteomic analysis was conducted. NSF-HA treatment was associated with much decreased co-IP of proteins. The authors suggest that CD44 is shielded by vHMM-HA whereas HMM-HA does not. Differential gene regulation in these studies demonstrated an enrichment of genes with a CD44-intracellular binding domain (CD44-ICD) motif in both HA-polymer length-dependent genes and p53 target genes. Thus NSF-HA suppressed HA polymer length-dependent p53 target genes. Further, the transcriptome of IMR90 cells after CD44 overexpression with and without NSF-HA demonstrated that NSF-HA treatment decreased the effect of CD44 overexpression on HA polymer length-dependent genes. Additionally, the phosphorylation of p53 was inhibited when cells were incubated with NSF-HA. Knockdown of p53 in IMR90 cells resulted in a loss of the protective nature of NSF-HA. The following arise from review of this manuscript.

1. The major editorial comment was that the manuscript did not have sufficiently developed mechanisms for the observations being reported. To this end, the authors have added CD44 IP experiments and CD44-mediated changes in signaling. However, these remain observational data and several key issues are not addressed at all in the manuscript.
2. For instance, p53 from the NMR differs from other species (PMID: 25172923) and this is not addressed and experimentally examined in these studies. Thus, using IMR90 cells to interrogate CD44 action in the human IMR90 cells is not valid as the p53 is different from the NMR. Further, both CD44 and RHAMM have been implicated in NMR HA signaling (also PMID: 25172923). The potential role of RHAMM in differential responses to vHMM-HA has not been explored in the current studies.
3. The authors examine possible protein:protein interactions of CD44 affected by NSF-HA. However, multiple other mechanisms have been proposed that would need to be explored for a thorough analysis of the mechanisms by which vHMM-HA regulates these observations. For example, the formation of molecular complexes (PMID: 29581896) and picket fences for organization of cell membrane proteins (PMID: 30209546) need to be evaluated. More direct analyses of receptor organization would also be needed such as FRET (PMID: 23118219) and atomic force microscopy (PMID: 30007620). Further, other HA-binding proteins such as versican could also be involved (PMID: 19164294).
4. Is it possible that vHMM-HA used in the experiments is simply affecting the ability of the CD44 antibody to bind the receptor?
5. Multiple transcriptomic and proteomic analyses are presented. However, no validation of these results is provided.
6. The tumor suppressor p53 also regulates autophagy where nuclear localization upregulates and cytoplasmic localization inhibits autophagy (PMID: 25896632). The authors should examine the localization of p53 with and without vHMM-HA to determine functional differences in p53.
7. Do IMR90 cells express CD44? CD44 mRNA and protein should be described for all cells used and the effects of vHMM-HA versus HMM-HA reported. The authors should also knockdown CD44 and demonstrate effects on vHMM-HA-mediated gene expression changes. Does p53 activity increase? Also, does vHMM-HA treatment affect cell surface localization of CD44 or variant expression?
8. To address the conundrum of blocking p53 and the anti-cancer phenotype of the NMR, the authors suggest that pALTINK4a/b is induced at the p16 locus. The authors should then knockdown this gene and demonstrate effects on the observations that they report for vHMM-HA.
9. If the authors propose that vHMM-HA is affecting partial p53 activity, then what activities of p53 does vHMM-HA not suppress?
10. The authors also report that the intracellular domain of CD44 is involved, but provide no data to support this.

Reviewer #3 (Remarks to the Author):

The paper entitled "Naked mole-rat hyaluronan exhibits superior cytoprotective properties by modulating the p53 pathway" by Takasugi and colleagues provide evidence that vHMM-HA (>6.1 MDa) from NMR skin fibroblasts (NSF-HA), but not HMM-HA (~1-2 MDa), protect cells by enhancing cellular stress resistance in a CD44-dependent manner. Mechanistically, the authors suggest that the cytoprotective effect involve p53 pathway as shown by the specific decreased phosphorylation of p53

at Ser9 as well as the modulation of p53 target genes.

The study follows up impressive findings by this research group on the particular properties of the extremely high molecular weight hyaluronan (over 5-fold larger than human or mouse hyaluronan) secreted by naked mole-rat fibroblasts. The concept is very interesting, the manuscript is well-written and the results are well presented. However, there are major concerns that should be considered by the authors:

- Although the experimental approaches used to support the results and concept of the study are convincing, it is not entirely clear what are the exact intracellular HA-CD44 signaling pathways involved in the modulation of p53 pathway responsible, as the authors suggest, for the cytoprotective effects of NSF-HA. Moreover, although the authors show a transcriptional effect on genes encoding the regulators and interactors of p53 using transcriptomic analysis, they do not focus in detail on the p53 pathway albeit the important observation of the specific decrease in the phosphorylation of p53 at Ser9 but not Ser15, Ser46, Thr81. This is crucial given the emphasis given by the authors on p53 pathway also in the title of the manuscript. At this point, the authors should clarify the aa residues examined since in text (page 13, line 257) they state "In addition, vHMM-HA decreases the phosphorylation of p53 at Ser9, but not at Ser20, Ser46, and Thr81" but in supplementary Fig. 5 they show phosphorylation levels at Ser15, Ser46, Thr81.
- Another issue that should be further clarified is the speculations by the authors that "... vHMM-HA and HMM-HA regulate CD44 signals in different ways" (page 7, lines 137-138) and "These results show that CD44 protein-protein interactions are promoted by HMM-HA but are suppressed by vHMM-HA, which might be due to the shielding of CD44 by very large vHMM-HA molecules OR by clustering multiple CD44 receptor molecules" (page 8, lines 152-155). These speculations are not supported experimentally in the present paper. The possibility of a different receptor clustering of vHMM-HA compared to HMM-HA should not be the case since HA molecules of substantial lower MM still have the ability to form clusters with multiple CD44 receptors. On the other hand, the proposed shielding of CD44 by vHMM-HA compared to HMM-HA should be further investigated by using for example higher concentrations of HMM-HA. This could determine if the observed effect is solely size-dependent or also concentration-dependent.
- In Fig. 2A, although vHMM-HA (>6.1 MDa) is present in NSF-HA preparations, the population of HA molecules appear very heterogeneous as evidenced by the broad distribution of sizes (ranging from 1 MDa up to >6.1 MDa) shown in gel electrophoresis. A similar heterogeneity is observed also for MSF-HA which ranges from ~0.5 MDa to ~6MDa. The overlapping in HA sizes between NSF and MSF introduce complexity to the effect observed by NSF-HA. The results would be more convincing and supportive to the size-dependent effect of NSF-HA if these preparations were separated to more defined sized HA populations by size-exclusion techniques (i.e. chromatography).
- The authors used IMR90 cell line to evaluate the relevance of NSF-HA in human cells. However, these cells are human primary lung fibroblasts and not skin fibroblasts as NSF and MSF fibroblasts. This might not be the suitable system for the appropriate interpretation of the results due to cell origin difference and, therefore, a human cell line of the same origin (i.e. human skin fibroblasts) should be used. Moreover, there are no any data regarding the hyaluronan-synthesizing capacity (amounts and size of HA) of IMR90 cells.
- The effect of HA on CD44 protein-protein interactions was analyzed after overexpression of CD44 in IMR90 cells. The overexpression of a protein, however, may result in its forced interaction(s) with additional proteins which otherwise do not interact with this protein. The authors should state on what are the levels of CD44 (standard form and/or isoform(s)) that are constitutively expressed by IMR90 cells and, maybe, perform co-immunoprecipitation experiments for the endogenous CD44.
- A more detailed schematic illustration that depicts the proposed mechanism whereby (NSF-HA) protects cells by enhancing cellular stress resistance (especially at the level of p53 pathway) should be included as a main (not supplementary) figure.

Point-by-point responses to the reviewers' comments

We are deeply thankful to the editor and reviewers for their valuable comments and constructive suggestions. We believe that our new experiments greatly enhance the value of our work. Changes in the text are indicated by blue font.

Reviewer #1 :

1. This paper describes a remarkably thorough set of experiments to define the route by which vHMW-HA from the naked mole rat (NMR) protects mammalian cells (of mouse, human or NMR origin) from a variety of toxic stresses. The authors discovered those protective properties, and here have made substantial progress in pursuit of their underlying mechanistic basis.

The key conclusions are (1.) that vHMW-HA (but not HMW-HA) masks the major HA receptor, CD44, and thus obstructs its protein-protein interactions (based on pull-down proteomics after cross-linking); (2.) that transcriptional effects of vHMW-HA are mediated via suppression of p53 target genes (demonstrated by analyses of transcriptomic data). In both cases, ample and appropriate controls were included, strengthening the conclusions drawn while excluding most alternative interpretations.

These are very important studies, because they indicate likely routes for stress-protective interventions that in principle could be quite effectively applied to humans (although they remain to be designed) and used prophylactically.

The statistical analyses appear valid, and actually are more sophisticated (and appropriate to the data being compared) than I see in most publications. While replications of experiments are rarely included when publishing scientific reports, the fact that very consistent results were obtained by multiple, independent approaches, by single-point assays vs. time courses, and in different cell lines (human IMR90 cells and mouse skin fibroblasts), indicates a high likelihood of reproducibility in other laboratories. Methodological details are certainly sufficient to enable replication.

Response-1:

Thank you very much for the comments.

2. The paragraphs added to the Discussion will be appreciated by many readers. Supplementary Figure 6 could serve as a graphical abstract; alternatively, it could be moved to the main body of the paper.

Response-2:

Thank you very much. We have moved original Supplementary Figure 6 to the main Figure 5.

Reviewer #2 :

This is a revised manuscript from Takasugi et al. that is examining the potential mechanisms of longevity and cancer resistance mediated by hyaluronan (HA) in the naked mole rat (NMR). They demonstrate that NMR skin fibroblasts (NSF) are more resistant to oxidative stress-mediated cell death than mouse skin fibroblasts (MSF). The effect of HA was not from direct scavenging of reactive oxygen species (ROS) as preincubation with NMR-HA protected later exposure to oxidative stress. Blockade of CD44 by an antibody abrogated the effect of NSF-HA. Comparison of very high molecular mass HA (vHMM-HA) to HMM-HA demonstrated a lower oxidative stress resistance for HMM-HA in human fetal fibroblasts (IMR90 cells). Partial fragmentation of vHMM-HA abrogated its protective properties. The authors respond to the criticism of a lack of mechanistic insight by evaluating the protein:protein interactions of CD44. After overexpression of CD44 in IMR90 cells with and without incubation with NSF-HA, fragmented NSF-HA or PBS, CD44 IP and proteomic analysis was conducted. NSF-HA treatment was associated with much decreased co-IP of proteins. The authors suggest that CD44 is shielded by vHMM-HA whereas HMM-HA does not. Differential gene regulation in these studies demonstrated an enrichment of genes with a CD44-intracellular binding domain (CD44-ICD) motif in both HA-polymer length-dependent genes and p53 target genes. Thus NSF-HA suppressed HA polymer length-dependent p53 target genes. Further, the transcriptome of IMR90 cells after CD44 overexpression with and without NSF-HA demonstrated that NSF-HA treatment decreased the effect of CD44 overexpression on HA polymer length-dependent genes. Additionally, the phosphorylation of p53 was inhibited when cells were incubated with NSF-HA. Knockdown of p53 in IMR90 cells resulted in a loss of the protective nature of NSF-HA. The following arise from review of this manuscript.

1. The major editorial comment was that the manuscript did not have sufficiently developed mechanisms for the observations being reported. To this end, the authors have added CD44 IP experiments and CD44-mediated changes in signaling. However, these remain observational data and several key issues are not addressed at all in the manuscript.

For instance, p53 from the NMR differs from other species (PMID: 25172923) and this is not addressed and experimentally examined in these studies. Thus, using IMR90 cells to interrogate CD44 action in the human IMR90 cells is not valid as the p53 is different from the NMR.

Response-1:

Our goal was to investigate the effect of vHMM-HA on human cells as a possible therapeutic strategy. To this end we used human cell line IMR90. Thus it is valid to use human IMR90

cells in this study. We agree, however, that it is useful for the readers to mention potential functional differences of p53 among human, mouse, and NMR. We discuss these points in the revised manuscript (page 13, line 265 to page 14, line 270).

Also, please note that our paper (Keane *et al.*, (2014) *Bioinformatics*. 30(24):3558-60) cited by the reviewer only suggests some positive selection of p53 amino acids but does not show any functional differences. Not surprisingly, amino acids sequences of p53 are not identical among species, not only between human and NMR.

2. Further, both CD44 and RHAMM have been implicated in NMR HA signaling (also PMID: 25172923). The potential role of RHAMM in differential responses to vHMM-HA has not been explored in the current studies.

Response-2:

To address this concern, we conducted knockdown experiments of CD44 and RHAMM, and found that CD44 siRNA but not RHAMM siRNA blocks the cytoprotective effect of NSF-HA (Supplementary Fig. 2; page 6, line 103-7).

3. The authors examine possible protein:protein interactions of CD44 affected by NSF-HA. However, multiple other mechanisms have been proposed that would need to be explored for a thorough analysis of the mechanisms by which vHMM-HA regulates these observations. For example, the formation of molecular complexes (PMID: 29581896) and picket fences for organization of cell membrane proteins (PMID: 30209546) need to be evaluated.

Response-3:

The paper cited by the reviewer regarding molecular complexes (PMID:29581896) investigated the complex formation among I κ B α , Erk, WWOX, Smad4, and Hyal2 using FRET. Another paper cited by the reviewer (PMID: 30209546) mentioned a study showing that CD44 can be immobilized and form “picket fence” upon binding to F-actin via its intracellular domain and thereby obstruct the diffusion of proximal Fc γ receptors.

While vHMM-HA might affect the formation of Hyal2 complex, CD44 picket fence, or any other pathway, there is no data to suggest that these phenomena are involved in the protective effect of vHMM-HA. We showed that CD44 and p53 play a critical role in the cytoprotective effect of vHMM-HA and that vHMM-HA and HMM-HA have contrasting effects on CD44 protein-protein interaction and CD44-dependent gene expression; However, we do not exclude the possibility that other HA-binding proteins, as well as other aspects of CD44, such as formation of picket fence, may be also involved in the cytoprotective effect or other functions of vHMM-HA. We agree that it would be useful for the readers to mention these possibilities and thus discussed these points in the revised manuscript (page 14, line

271-78). However, these issues are clearly beyond the scope of this study. Please understand that we cannot test the effects of vHMM-HA on all HA- and CD44-related phenomena.

4. More direct analyses of receptor organization would also be needed such as FRET (PMID: 23118219) and atomic force microscopy (PMID: 30007620).

Response-4:

In accordance with the reviewer's comment, we further analyzed CD44 organization. Instead of conducting FRET (which we tried but was technically difficult), we simultaneously overexpressed CD44-FLAG and CD44-YFP in IMR90 cells and examined their interaction by FLAG immunoprecipitation followed by Western blot analysis. Consistent with our mass-spec data, cNSF-HA reduced the amount of co-immunoprecipitated CD44-YFP, whereas fNSF-HA increased it (Fig. 2H, I; page 8, line 157 to page 9, line 163).

5. Further, other HA-binding proteins such as versican could also be involved (PMID: 19164294).

Response-5:

We showed that CD44 and p53 play a critical role in the cytoprotective effect of vHMM-HA. However, we do not exclude the possibility that other proteins are also important, as discussed in the revised manuscript (page 14, line 271-78). Please understand that this issue is beyond the scope of this study.

6. Is it possible that vHMM-HA used in the experiments is simply affecting the ability of the CD44 antibody to bind the receptor?

Response-6:

The binding of CD44 antibody to CD44 is not significantly affected by vHMM-HA, since the amount of immunoprecipitated CD44 was not significantly different among PBS-, HMM-HA-, and vHMM-HA-treated cells (Supplementary Dataset S1). Moreover, in our CD44-IP experiment, we normalized the amount of co-immunoprecipitated proteins to the amount of immunoprecipitated CD44. Therefore, our conclusion is valid even if the binding of CD44 antibody to CD44 was slightly affected by vHMM-HA.

7. Multiple transcriptomic and proteomic analyses are presented. However, no validation of these results is provided.

Response-7:

We performed additional experiments in order to validate our omics analyses. The polymer length-dependent effects of HA on the expression of p53 target genes (RRM2B, PRRX2, and

PYCARD) were confirmed by RT-qPCR (Supplementary Fig. 8; page 11, line 220-3). The polymer length-dependent effect of HA on CD44 protein-protein interaction was confirmed by co-immunoprecipitation followed by Western blot (Fig. 2H, I; page 8, line 157 to page 9, line 163).

8. The tumor suppressor p53 also regulates autophagy where nuclear localization upregulates and cytoplasmic localization inhibits autophagy (PMID: 25896632). The authors should examine the localization of p53 with and without vHMM-HA to determine functional differences in p53.

Response-8:

We tested the localization of p53, and found that HMM-HA and vHMM-HA do not affect nuclear and cytoplasmic p53 levels (Supplementary Fig. 7D-G; page 10, line 202 to page 11, line 203).

9. Do IMR90 cells express CD44? CD44 mRNA and protein should be described for all cells used and the effects of vHMM-HA versus HMM-HA reported. The authors should also knockdown CD44 and demonstrate effects on vHMM-HA-mediated gene expression changes. Does p53 activity increase? Also, does vHMM-HA treatment affect cell surface localization of CD44 or variant expression?

Response-9:

We showed that the cells used in this study express CD44 in both mRNA and protein levels (Supplementary Fig. 4B, C; page 8, line 143-4). Almost all CD44 was standard form and there was little to no expression of CD44 variants regardless of HA treatment. We showed that the polymer length-dependent effects of HA on the expression of p53 target genes are blocked by CD44 knockdown (Supplementary Fig. 8; page 11, line 220-3). We also showed the effects of HMM-HA and vHMM-HA on CD44-CD44 interaction (as mentioned in Response-4) and CD44-ICD levels (as mentioned in Response-12).

10. To address the conundrum of blocking p53 and the anti-cancer phenotype of the NMR, the authors suggest that pALTINK4a/b is induced at the p16 locus. The authors should then knockdown this gene and demonstrate effects on the observations that they report for vHMM-HA.

Response-10:

The focus of this manuscript is on cytoprotective effect of vHMM-HA on human cells. Please note that, unlike NMR, human cells do not express pALTINK4a/b nor go into cell cycle arrest upon vHMM-HA treatment (Supplementary Fig. 1C; page 6, line 99-102). We removed the

former discussion from the manuscript. Rather, we now discuss that the vHMM-HA-induced partial attenuation of p53 may not necessarily compromise its tumor suppressor activity (page 15, line 288-94). Indeed, as a whole, p53 target genes respond very similarly to oxidative stress in PBS- and HA-incubated cells (Supplementary Fig. 7C; page 10, line 201-2). Differential transcriptional modulation of a subset of p53 target genes by cNSF-HA and fNSF-HA could be a consequence of altered expression of genes encoding regulators and interactors of p53 that are regulated by CD44-regulated transcription factors and CD44-ICD.

11. If the authors propose that vHMM-HA is affecting partial p53 activity, then what activities of p53 does vHMM-HA not suppress?

Response-11:

In the revised manuscript, we showed that majority of the p53 target genes respond similarly to the oxidative stress in cNSF-HA, fNSF-HA, and PBS-incubated cells (Supplementary Fig. 7C; page 10, line 201-2). We found that target genes of CD44-ICD and CD44-regulated transcription factors are enriched among HA polymer length-dependent genes that are upstream of p53, but not among downstream targets of p53 (Fig. 3D; page 11, line 203-8). This suggest that HA polymer length-dependent regulation of the subset of p53 target genes is a consequence of altered expression of genes upstream of p53 that are regulated by CD44-ICD and CD44-regulated transcription factors. We confirmed that at least some of the HA polymer length-dependent p53 target genes are regulated in a CD44-dependent manner (Supplementary Fig. 8; page 11, line 220-3).

12. The authors also report that the intracellular domain of CD44 is involved, but provide no data to support this.

Response-12:

To address this concern, we conducted Western blot and found that vHMM-HA and HMM-HA differently affect CD44-ICD levels (Supplementary Fig. 6; page 10, line 191-3).

Reviewer #3 :

1. The paper entitled “Naked mole-rat hyaluronan exhibits superior cytoprotective properties by modulating the p53 pathway” by Takasugi and colleagues provide evidence that vHMM-HA (>6.1 MDa) from NMR skin fibroblasts (NSF-HA), but not HMM-HA (~1-2 MDa), protect cells by enhancing cellular stress resistance in a CD44-dependent manner. Mechanistically, the authors suggest that the cytoprotective effect involve p53 pathway as shown by the specific decreased phosphorylation of p53 at Ser9 as well as the modulation of p53 target genes.

The study follows up impressive findings by this research group on the particular properties of the extremely high molecular weight hyaluronan (over 5-fold larger than human or mouse hyaluronan) secreted by naked mole-rat fibroblasts. The concept is very interesting, the manuscript is well-written and the results are well presented.

Response-1:

Thank you very much for the comment.

2 However, there are major concerns that should be considered by the authors: Although the experimental approaches used to support the results and concept of the study are convincing, it is not entirely clear what are the exact intracellular HA-CD44 signaling pathways involved in the modulation of p53 pathway responsible, as the authors suggest, for the cytoprotective effects of NSF-HA. Moreover, although the authors show a transcriptional effect on genes encoding the regulators and interactors of p53 using transcriptomic analysis, they do not focus in detail on the p53 pathway albeit the important observation of the specific decrease in the phosphorylation of p53 at Ser9 but not Ser15, Ser46, Thr81. This is crucial given the emphasis given by the authors on p53 pathway also in the title of the manuscript. At this point, the authors should clarify the aa residues examined since in text (page 13, line 257) they state “In addition, vHMM-HA decreases the phosphorylation of p53 at Ser9, but not at Ser20, Ser46, and Thr81” but in supplementary Fig. 5 they show phosphorylation levels at Ser15, Ser46, Thr81.

Response-2:

We sincerely apologize for the typo. We measured the phosphorylation of p53 on Ser15, not Ser20. In accordance with the reviewer’s comment, we further investigated the role of p53 and show that overall p53 activities as well as nuclear and cytoplasmic p53 levels are largely unaffected by HA (Supplementary Fig. 7C-G; page 10, line 201 to page 11, line 204). We found that target genes of CD44-ICD (CD44 intra-cytoplasmic domain) and CD44-regulated transcription factors are enriched among HA polymer length-dependent genes that are upstream of p53, but not among downstream targets of p53 (Fig. 3D; page 11, line 203-8). This suggests

that HA polymer length-dependent regulation of the subset of p53 target genes is a consequence of altered expression of genes upstream of p53 that are regulated by CD44-ICD and CD44-regulated transcription factors. We confirmed that at least some of the HA polymer length-dependent p53 target genes are regulated in a CD44-dependent manner (Supplementary Fig. 8; page 11, line 220-3). We also confirmed that vHMM-HA and HMM-HA differently affect CD44-ICD levels (Supplementary Fig. 6; page 10, line 191-3). Regardless of these additional findings, we changed the title of this manuscript from “Naked mole-rat hyaluronan exhibits superior cytoprotective properties by modulating the p53 pathway” to “Naked mole-rat very-high-molecular-mass hyaluronan (vHMM-HA) exhibits superior cytoprotective properties than the shorter HMM-HA”, in order to emphasize the most important finding of this study.

3. Another issue that should be further clarified is the speculations by the authors that “... vHMM-HA and HMM-HA regulate CD44 signals in different ways” (page 7, lines 137-138) and “These results show that CD44 protein-protein interactions are promoted by HMM-HA but are suppressed by vHMM-HA, which might be due to the shielding of CD44 by very large vHMM-HA molecules OR by clustering multiple CD44 receptor molecules” (page 8, lines 152-155). These speculations are not supported experimentally in the present paper. The possibility of a different receptor clustering of vHMM-HA compared to HMM-HA should not be the case since HA molecules of substantial lower MM still have the ability to form clusters with multiple CD44 receptors. On the other hand, the proposed shielding of CD44 by vHMM-HA compared to HMM-HA should be further investigated by using for example higher concentrations of HMM-HA. This could determine if the observed effect is solely size-dependent or also concentration-dependent.

Response-3:

We thank the reviewer for this comment. We now provide additional evidence that vHMM-HA shields CD44 from protein-protein interactions. In order to clarify the effect of vHMM-HA on CD44-CD44 interaction, we simultaneously overexpressed CD44-FLAG and CD44-YFP in IMR90 cells and examined their interaction by FLAG immunoprecipitation followed by Western blot analysis. We found that cNSF-HA reduced the amount of co-immunoprecipitated CD44-YFP, whereas fNSF-HA increased it (Fig. 2H, I; page 8, line 157 to page 9, line 163). This result suggests that very large vHMM-HA molecules shield CD44 and reduced its interaction with other proteins. In addition, we showed that 1 MDa HMM-HA (Select-HA™) does not protect cells from oxidative stress even at higher concentration (5-fold higher concentration than that of vHMM-HA used in our study) (Supplementary Fig. 3H-I; page 7, line 131-4).

4. In Fig. 2A, although vHMM-HA (>6.1 MDa) is present in NSF-HA preparations, the

population of HA molecules appear very heterogeneous as evidenced by the broad distribution of sizes (ranging from 1 MDa up to >6.1 MDa) shown in gel electrophoresis. A similar heterogeneity is observed also for MSF-HA which ranges from ~0.5 MDa to ~6MDa. The overlapping in HA sizes between NSF and MSF introduce complexity to the effect observed by NSF-HA. The results would be more convincing and supportive to the size-dependent effect of NSF-HA if these preparations were separated to more defined sized HA populations by size-exclusion techniques (i.e. chromatography).

Response-4:

We thank the reviewer for this comment. We addressed the issue of HA polydispersity by using gel-extracted vHMM-HA and synthetic hyaluronan (Select-HA™) with a uniform molecular mass of 1 MDa. Gel-extracted vHMM-HA, but not Select-HA™ protected IMR90 cells from oxidative stress (Supplementary Fig. 3F-I; page 7, line 131-4). Thus we can now conclude that vHMM-HA larger than 6.1 MDa has superior protective effect compared to 1 MDa HA or MSF-HA that consists of HMM-HA smaller than 6.1 MDa (page 14, line 279-86).

5. The authors used IMR90 cell line to evaluate the relevance of NSF-HA in human cells. However, these cells are human primary lung fibroblasts and not skin fibroblasts as NSF and MSF fibroblasts. This might not be the suitable system for the appropriate interpretation of the results due to cell origin difference and, therefore, a human cell line of the same origin (i.e. human skin fibroblasts) should be used. Moreover, there are no any data regarding the hyaluronan-synthesizing capacity (amounts and size of HA) of IMR90 cells.

Response-5:

Our human skin fibroblast cell line HCA2 (Gorbunova *et al.*, (2002) *J Biol Chem.* 11:277(41):38540-9) seems to have high HA degrading activity, as shown in the figure below. Therefore this cell line seems to be unsuitable for evaluating the effect of vHMM-HA. Rather, we decided to use IMR90 cells since this is the most studied primary human cell line in regard of cellular stress response. In accordance with the reviewer's comment, we showed the hyaluronan-synthesizing capacity of IMR90 cells in the manuscript (Supplementary Fig. 3A). Please also note that we do not intend to compare cells of different species.

Culture supernatant of human skin fibroblast HCA2 cells contain shorter HA than that of IMR90 cells.

6. The effect of HA on CD44 protein-protein interactions was analyzed after overexpression of CD44 in IMR90 cells. The overexpression of a protein, however, may result in its forced interaction(s) with additional proteins which otherwise do not interact with this protein. The authors should state on what are the levels of CD44 (standard form and/or isoform(s)) that are constitutively expressed by IMR90 cells and, maybe, perform co-immunoprecipitation experiments for the endogenous CD44.

Response-6:

To address the reviewer's comment, we examined the expression levels of endogenous and exogenous CD44 (standard and isoforms) (Supplementary Fig. 4B, C; page 8, line 143-4). We show that the standard form isoform is predominantly expressed and is not affected by HA length. We also mention the potential artefacts that can be caused by CD44 overexpression (page 9, line 181 to page 10, line 184).

7. A more detailed schematic illustration that depicts the proposed mechanism whereby (NSF-HA) protects cells by enhancing cellular stress resistance (especially at the level of p53 pathway) should be included as a main (not supplementary) figure.

Response-7:

In accordance with the reviewer's comment, we modified the schematic illustration and included in Fig. 5.

REVIEWERS' COMMENTS:

Reviewer #2 (Remarks to the Author):

The authors have adequately addressed all the previous critiques.

Reviewer #3 (Remarks to the Author):

The paper entitled "Naked mole-rat very-high-molecular-mass hyaluronan (vHMM-HA) exhibits superior cytoprotective properties than the shorter HMM-HA" by Takasugi and colleagues provide evidence that vHMM-HA (>6.1 MDa) from NMR skin fibroblasts (NSF-HA), but not HMM-HA (~1-2 MDa), protect cells by enhancing cellular stress resistance in a CD44-dependent manner.

Mechanistically, the authors suggest that the cytoprotective effect involve p53 pathway as shown by the specific decreased phosphorylation of p53 at Ser9 as well as the modulation of p53 target genes. As stated in my previous report, the study follows up impressive findings by this research group on the particular properties of the extremely high molecular weight hyaluronan (over 5-fold larger than human or mouse hyaluronan) secreted by naked mole-rat fibroblasts. The concept is very interesting, the manuscript is well-written and the results are well presented.

In my opinion, the authors adequately addressed the concerns raised in the original submission. The new experiments have strengthen the conclusions while the authors have changed the title of the ms to "Naked mole-rat very-high-molecular-mass hyaluronan (vHMM-HA) exhibits superior cytoprotective properties than the shorter HMM-HA" to emphasize the most important findings of the study. I think the following title would be also appropriate: "Naked mole-rat very-high-molecular-mass hyaluronan exhibits superior cytoprotective properties".